

# Conditions for the occurrence of seismic sequences in a fault system

Michele Dragoni[1] and Emanuele Lorenzano[1]

[1]Alma Mater Studiorum Università di Bologna , Dipartimento di Fisica e Astronomia, Viale Carlo Berti Pichat 8, 40127 Bologna, Italy

*Correspondence to:* Michele Dragoni (michele.dragoni@unibo.it), Emanuele Lorenzano (emanuele.lorenzano2@unibo.it)

**Abstract.** We consider a system made of $n$ coplanar faults and study the conditions under which it may originate a sequence of $n$ seismic events of similar magnitudes. The system is placed in an elastic half-space and is subject to a constant and uniform strain rate by tectonic motions. The state of the system is described by $n$ variables that are the Coulomb stresses of the faults. Coulomb stresses increase steadily during the interseismic intervals and change suddenly in connection with seismic

events. A permutation of the first $n$ integers can be associated with each state of the system, expressing the order of Coulomb stresses according to their magnitudes. The permutation controls the evolution of the system. The state space can be divided into $n!$ subsets corresponding to each of the $n!$ permutations. The representative point of the system moves continuously in one of the subsets during the interseismic intervals and switches to a different subset in connection with seismic events. The order of events in a sequence can be also expressed as a permutation of the first $n$ integers, so that the number of possible

sequences is equal to $n!$. The order of events is determined by the initial stress state, the stress drops and the stress transfers associated with each event. If the sequence is made of $n$ similar events and the differences between Coulomb stresses are always greater than the mutual stress transfers, the order is given by the initial permutation. However, the order implicit in the initial permutation is generally modified during the sequence, due to changes in the differences between Coulomb stresses and to different stress drops of the events. The model allows the retrieval of the stress states of a fault system from the observation

of a seismic sequence. As an example, the model is applied to the 2012 Emilia (Italy) seismic sequence and enlightens the complex interplay between the fault dislocations that produced the observed order of events.

## 1   Introduction

Seismic sequences are a characteristic aspect of seismic phenomenology. Recent examples in Italy are the 1997-98 Umbria-Marche sequence (Morelli et al., 2000; Salvi et al., 2000; Santini et al., 2004) and the 2012 Emilia sequence (Malagnini et al.,

2012; Scognamiglio et al., 2012; Castro et al., 2013; Pezzo et al., 2013). We call "seismic sequence" a series of earthquakes generated by sources located in a relatively small region (in the order of 100 km) and occurring in a time interval (in the order of few months) much shorter than the intervals during which the system is at rest. The interval elapsing between two seismic sequences in the same region is in the order of several decades at least. We do not include in this definition aftershock sequences following a greater event, that may have similar features but are strongly conditioned by the main shock.



Sequences are originated by fault systems that produce similar earthquakes as to mechanism and magnitude. A sequence is typically made of a small number ($< 10$) of larger events having a medium magnitude, in general between 5 and 6, plus a greater number of smaller events. We take into account the larger events, neglecting the smaller ones.

Since the faults of the system are close to each other, it is believed that fault interaction plays a major role in seismic sequences. Fault interaction and its role in earthquake triggering have been widely studied (e.g. Steacy et al., 2005). Whenever a fault slips, it transfers stress to neighbouring faults, thus anticipating the instant of time when they slip in their turn. Therefore the interaction has the effect of concentrating the events in a shorter time interval, hence reducing the duration of the sequence (e.g. Tallarico et al., 2005).

In the present paper, we consider a model of a fault system and investigate the conditions under which the system may originate seismic sequences with the characteristics described above. In particular, we ask the following questions: 1) which are the stress conditions under which a sequence may take place? 2) what determines the order of seismic events in the sequence? 3) what makes the order of events change from one sequence to the following one? 4) is the observed order of events informative about the state of stress before and after the sequence?

The model is applied to the 2012 Emilia (Italy) seismic sequence, that was made of seven major events with similar focal mechanisms and with magnitudes between 5 and 6, occurred in a time interval of 15 days.

## 2 The model

We consider a system made of $n$ plane faults, that we assume to be coplanar and lined up, with the same strike and dip angles (Fig. 1). The fault system is placed in an elastic half-space with Lamé constants $\lambda$ and $\mu$. We introduce a coordinate system ($x$, $y$, $z$) such that the $x$ axis coincides with fault strike, $y$ is the horizontal direction perpendicular to strike and $z$ is depth. Let $\delta$ be the dip angle of the faults.

We number the faults from 1 to $n$, starting from one end of the system. Let $A_i$ be the area of the $i$-th fault and $r_{ij}$ be the distance between the centres of the $i$-th and the $j$-th fault. We introduce the following assumptions:

1) the fault system is subject to a strain rate $\dot{e}$, that is constant in time and uniform in space;

2) the onset of seismic events is controlled by the average values of tangential traction and static friction on fault surfaces;

3) fault slip is a step function of time and does not produce overshooting;

4) each fault slips once and only once during a sequence;

5) there is no simultaneous slip of two or more faults and a finite time interval elapses between the failures of any two faults;

6) the duration of a sequence is much shorter than the interval between two consecutive sequences;

7) the system is not subject to external stress perturbations.

Assumption 1 is reasonable, since by definition the $n$ faults belong to the same seismogenic region, for which the same tectonic mechanism is observed. Assumptions 2 and 3 are based on the fact that we are not interested in the details of each event, that has a much shorter duration than the duration of the sequence, but rather in the relationship between the $n$ events. Assumptions 4, 5 and 6 are dictated by the characteristics of the sequences we are describing. As to assumption 7, it is a fact





that the evolution of a fault system can be altered by external perturbations. Any fault system is not isolated, but is surrounded by other faults, that may transfer stress to it whenever they slip (e.g. Dragoni and Piombo, 2015). Generally, contributions from external faults may be numerous during an interseismic interval, but they are smaller than contributions from faults belonging to the system, due to greater distances and to different orientations of fault surfaces. Such contributions may also partially

cancel each other.

In the case of normal and reverse faults, we assume plane strain, according to the Anderson model (Anderson, 1951; Sibson, 1974; Turcotte and Schubert, 2002). The nonvanishing strain components are

$$e_{yy} = \dot{e}t, \qquad e_{zz} = -\frac{\lambda}{\lambda + 2\mu} e_{yy} \tag{1}$$

where $\dot{e}$ is positive for tensile strain and negative for compressive strain. The stress components are

$$\sigma_{xx} = \nu \sigma_{yy}, \qquad \sigma_{yy} = \frac{2\mu}{1-\nu} e_{yy} \tag{2}$$

where $\nu$ is the Poisson modulus. We introduce the stress rate

$$\dot{\sigma} = \frac{2\mu}{1-\nu} \dot{e} \tag{3}$$

The rates of normal and tangential traction on the faults are then

$$\dot{\sigma}_n = -\frac{\dot{\sigma}}{2}(1 - \cos 2\delta), \quad \dot{\sigma}_t = \pm \frac{\dot{\sigma}}{2} \sin 2\delta \tag{4}$$

where the upper sign in $\sigma_t$ is for normal faults and the lower sign is for reverse faults. In the case of transcurrent faults, we consider simple shear, with strain and stress components

$$e_{xy} = \dot{e}t, \qquad \sigma_{xy} = 2\mu e_{xy} \tag{5}$$

In this case, we define

$$\dot{\sigma} = 2\mu \dot{e} \tag{6}$$

and the rates of normal and tangential traction on the faults are

$$\dot{\sigma}_n = 0, \quad \dot{\sigma}_t = \dot{\sigma} \tag{7}$$

Let $\sigma_i$ be the average tangential traction applied to the $i$-th fault in the slip direction and $\tau_i$ be the average static friction of the $i$-th fault. We define the Coulomb stress (Stein, 1999) of the $i$-th fault as

$$x_i = \sigma_i - \tau_i, \quad i = 1, 2, \ldots n \tag{8}$$

Since the $\sigma_i$ are always positive or zero, the $x_i$ range between $-\tau_i$ and zero. When $x_i = 0$, an earthquake is generated by the $i$-th fault. The rates of $\sigma_i$ and $\tau_i$ are respectively

$$\dot{\sigma}_i = \dot{\sigma}_t, \qquad \dot{\tau}_i = \kappa \dot{\sigma}_n \tag{9}$$





where $\kappa$ is the coefficient of static friction. Then the rate of Coulomb stress is

$$\dot{x} = k\dot{\sigma} \tag{10}$$

where

$$k = \sin\delta(\kappa\sin\delta \pm \cos\delta) \tag{11}$$

for normal and reverse faults and

$$k = 1 \tag{12}$$

for transcurrent faults. Then, in the absence of earthquakes, the Coulomb stress of the $i$-th fault changes in time as

$$x_i(t) = x_{0i} + \dot{x}t \tag{13}$$

where $x_{0i}$ is the Coulomb stress at an arbitrary time $t = 0$.

Due to the presence of friction, the set of $n$ faults is a nonlinear dynamical system. The Coulomb stresses $x_i$ can be considered as the components of an $n$-dimensional vector $\mathbf{x}(t)$ describing the state of the system as a function of time. The possible states of the system belong to an $n$-dimensional parallelepiped $S$, defined by the $n$ disequalities

$$-\tau_i \le x_i \le 0 \tag{14}$$

According to assumption 5, all the components of $\mathbf{x}$ are different from each other. Therefore one (and only one) component
will vanish first, generating the first event in the sequence. Whenever an earthquake occurs, the fault dislocation produces a static stress field that is transferred to the system and modifies the Coulomb stress of all faults, producing a sudden change in $\mathbf{x}$.

In general, the change in Coulomb stress on the $j$-th fault due to the failure of the $i$-th fault can be written as

$$\Delta x_{ij}(t) = \Delta\sigma_{ij}H(t) + \Delta\sigma'_{ij}(t) \tag{15}$$

where $\Delta\sigma_{ij}$ is the coseismic change in tangential traction, $H$ is the Heaviside function and $\Delta\sigma'_{ij}$ is the change in traction due to time-dependent processes, including pore fluid diffusion, afterslip and viscoelastic relaxation. Since we have assumed that faults are coplanar, there are no changes in normal stress on the fault plane.

The traction $\Delta\sigma_{ij}$ could be calculated from the formulae for a rectangular dislocation in an elastic half-space collected by Okada (1992). However, if $r_{ij} > 1.5\sqrt{A_i}$, the traction of a finite dislocation source is virtually indistinguishable from that of a
point-like double-couple source in an unbounded medium and the latter simpler formula can be used (Appendix A).

A Poisson solid ($\lambda = \mu$) is considered. Accordingly, if $m_i$ is the seismic moment of the $i$-th fault, we have

$$\Delta\sigma_{ij} = \frac{5m_i}{12\pi r_{ij}^3}, \quad i \ne j \tag{16}$$


in the case of a strike-slip mechanism and

$$\Delta\sigma_{ij} = \frac{m_i}{6\pi r_{ij}^3}, \quad i \neq j \tag{17}$$

in the case of a dip-slip mechanism. As to the stress change of the $i$-th fault, it is

$$\Delta\sigma_{ii} = -\Delta\sigma_i \tag{18}$$

where $\Delta\sigma_i$ is the static stress drop, that can be estimated from the average slip $u_i$ and the fault area $A_i$ as

$$\Delta\sigma_i = C\frac{\mu u_i}{\sqrt{A_i}} \tag{19}$$

where $C$ is a nondimensional constant of the order of unity determined by the geometry of the fault (Kanamori, 2001). According to discrete fault models (e.g. Dragoni and Piombo, 2015), the stress drop is a fraction

$$f = 2(1-\epsilon) \tag{20}$$

of static friction, where $\epsilon$ is the ratio between the average values of dynamic and static frictions, that we assume to be the same for all faults.

If the medium is porous and saturated with fluids, the coseismic stress field induces a fluid flow that changes the stress field in turn (e.g. Wang, 2000; Piombo et al., 2005). As shown in Appendix B, the effect of fluid diffusion is at least one order of magnitude smaller than coseismic stress transfer: for the sake of simplicity, we do not consider it in the following.

As to afterslip and viscoelastic relaxation, the events we are considering are relatively small and it is assumed that they do not produce appreciable afterslip nor impose considerable stress to deeper ductile regions that may relax it afterwards. Viscoelastic relaxation of lithospheric rocks may change the stress distribution in the long term as a consequence of larger earthquakes (e.g. Dragoni and Lorenzano, 2015).

## 3 Evolution of the system

Let $t_k$ be the occurrence times of the events in the sequence ($k = 1, 2, \dots n$), so that the durations of the interseismic intervals are

$$\Delta t_k = t_{k+1} - t_k, \quad k = 1, 2, \dots n-1 \tag{21}$$

Then the initial state is $\mathbf{x}(t_1-)$. If the first event is due to the failure of the $i_1$-th fault, $\mathbf{x}$ has a sudden change and its $k$ component becomes

$$x_k(t_1+) = x_k(t_1-) + \Delta x_{i_1 k} \tag{22}$$

Afterwards, $\mathbf{x}$ changes continuously in time, as a consequence of tectonic loading, according to

$$x_k(t) = x_k(t_1+) + \dot{x}(t - t_1) \tag{23}$$





At $t = t_2$, the second event takes place, due to the failure of the $i_2$-th fault, so that $\mathbf{x}$ has another sudden change, and so on. At the end of the sequence, the state is

$$x_k(t_n+) = x_k(t_n-) + \Delta x_{i_n k} \tag{24}$$

that can be written as

$$x_k(t_n+) = x_k(t_1-) + \dot{x}\,\Delta t + \sum_{j=1}^{n} \Delta x_{jk} \tag{25}$$

where

$$\Delta t = \sum_{k=1}^{n-1} \Delta t_k = t_n - t_1 \tag{26}$$

is the duration of the sequence, that can be written as

$$\Delta t = -\frac{x_{i_n}(t_1-)}{\dot{x}} - \frac{1}{\dot{x}} \sum_{j=1}^{n} \Delta x_{i_j i_n}, \quad i_j \neq i_n \tag{27}$$

In Eq. (25) the difference between the final and the initial state is made of two terms: the first one is tectonic loading during the time interval $\Delta t$; the second one is the effect of earthquakes. The latter term has the effect of concentrating in a shorter time interval a series of events that otherwise would be farther in time. The shortening in duration is obtained by calculating how much the instant $t_n$ of the last event is anticipated. The decrease in $t_n$ is due to the sum of the stresses that are transferred to the $i_n$-th fault from the other $n-1$ faults. From Eq. (27), the duration of the sequence in the absence of interaction is

$$\Delta t' = \Delta t + \frac{1}{\dot{x}} \sum_{j=1}^{n} \Delta x_{i_j i_n}, \quad i_j \neq i_n \tag{28}$$

The interseismic intervals (21) can be calculated as

$$\Delta t_k = -\frac{x_{i_{k+1}}(t_k+)}{\dot{x}}, \quad k = 1, 2, \dots n-1 \tag{29}$$

During the interseismic intervals, the representative point $\mathbf{x}$ moves along a line defined by the parametric equations (13), that is parallel to the line

$$x_1 = x_2 = \cdots = x_n \tag{30}$$

Thanks to a rotation $\mathbf{R}$, the coordinate system $(x_1, \dots x_n)$ can be changed into a system $(\xi_1, \dots \xi_n)$ such that the $\xi_n$ axis coincides with line (30). Hence the evolution of the system can be more easily represented in the $(n-1)$-dimensional hyperplane $\xi_n = 0$. An example will be shown in section 7 for $n = 3$.





## 4 Retrieval of the initial and final states

On the basis of the model, if we observe a seismic sequence, we can retrieve the state vector $\mathbf{x}(t)$ at any time during the sequence. In particular, we can calculate the state of the system at the beginning and at the end of the sequence.

Suppose that we observe a sequence made of $n$ events that can be ascribed to the failure of $n$ faults belonging to the same
system. Let $t_1$, $t_2$, ... $t_n$ be the observed occurrence times of the events. From the knowledge of the fault geometry and of the seismic moments, we can calculate the stress transfer matrix $\Delta x_{ij}$. If we know the strain rate $\dot{e}$ from geodetic measurements, we can calculate the stress rate $\dot{x}$ from Eq. (10).

Consider the generic fault $i_k$, that has produced the $k$-th event in the sequence. It is easy to see that the Coulomb stress of fault $i_k$ at the beginning of the sequence is

$$x_{i_k}(t_1-) = -\dot{x}(t_k - t_1) - \sum_{j=1}^{k-1} \Delta x_{i_j i_k} \tag{31}$$

Apart from the signs, the first term in the rhs is the stress accumulated on the fault from the beginning of the sequence up to instant $t_k$ and the second term is the sum of stress transfers that fault $i_k$ has received from faults $i_1$, $i_2$, ... $i_{k-1}$ that slipped before it. Hence, apart from the sign, the rhs is the total stress accumulated on fault $i_k$ since the beginning of the sequence. This stress must cancel the initial Coulomb stress $x_{i_k}(t_1-)$: hence the initial Coulomb stress must be the opposite of the
accumulated stress.

As to the final state of fault $i_k$, it is given by Eq. (25). If we replace $x_{i_k}(t_1-)$ in Eq. (25) with its expression (31), we obtain

$$x_{i_k}(t_n+) = \dot{x}(t_n - t_k) + \sum_{j=k}^{n} \Delta x_{i_j i_k} \tag{32}$$

Since the Coulomb stress of fault $i_k$ was equal to zero at $t = t_k-$, the final stress is equal to the tectonic stress accumulated in the time interval from $t_k$ to $t_n$ plus the stress drop associated with the failure of fault $i_k$ and the stress transfers of the faults
$i_{k+1}$, $i_{k+2}$, ... $i_n$ that have slipped after fault $i_k$.

Hence the initial Coulomb stress of fault $i_k$ depends only on what happened before the instant $t_k$, while the final Coulomb stress depends only on what happened after $t_k$. However, the retrieval of the complete state vector requires the knowledge of the entire sequence. In section 8, we shall retrieve the initial and final states of a fault system in a real case.

The degree of heterogeneity of the $x_i$ can be expressed by their standard deviation

$$s = \left[ \frac{1}{n} \sum_{i=1}^{n} (x_i - \bar{x})^2 \right]^{1/2} \tag{33}$$

where

$$\bar{x} = \frac{1}{n} \sum_{i=1}^{n} x_i \tag{34}$$

A relevant point for the subsequent evolution is whether the differences between the $x_i$ change during a sequence. We define

$$d_{ij}(t) = x_i(t) - x_j(t) \tag{35}$$





The $d_{ij}$ form an antisymmetric matrix having $n(n-1)$ nonvanishing components that are related by $(n-1)^2$ equations. Therefore $d_{ij}$ is known if we know only $n-1$ components, for example the $d_{1j}$ with $j = 2,3,\ldots n$. Thanks to Eq. (32), we obtain

$$d_{ij}(t_n+) - d_{ij}(t_1-) = \sum_{k=1}^{n}(\Delta x_{ki} - \Delta x_{kj}) \tag{36}$$

where the rhs is different from zero because the sum of stress transfers received by a fault during the sequence is in general different from that received by the other faults. In particular, faults located at the center of the system receive a greater total stress than faults located at the ends, if the events have similar magnitudes. For instance, if $n = 3$, fault 2 receives a greater stress transfer than faults 1 and 3.

## 5 The order of events

Since the components $x_i$ of the state vector are always different from each other, they can be ordered according to their magnitudes. Then, at any instant $t$ of time, the set $X$ of the $x_i(t)$ is a well-ordered set. This order controls the order of events in the seismic sequence.

Let $N_n$ be the set of the first $n$ natural numbers. With each state $\mathbf{x}$ of the system we can associate a permutation $\alpha$ of $N_n$, expressing the order of faults in relation to the value of their Coulomb stress:

$$\alpha = \begin{pmatrix} 1 & 2 & \ldots & n \\ i_1 & i_2 & \ldots & i_n \end{pmatrix} \tag{37}$$

so that

$$x_{i_1} = \max X \tag{38}$$

$$x_{i_k} = \max(X - \{x_{i_1}, x_{i_2}, \ldots x_{i_{k-1}}\}) \tag{39}$$

with $k = 2,3,\ldots n$. Hence the parallelepiped $S$ can be divided into a number $n!$ of subsets $S_j$ corresponding to the $n!$ permutations of $N_n$. During the interseismic intervals, the permutation $\alpha_j$ associated with the system does not change, because all the $x_i$ increase with the same rate, according to Eq. (13). Therefore $\mathbf{x}$ remains in the same subset $S_j$. However, when an event occurs, $\mathbf{x}$ switches to a different subset $S_k$, characterized by a permutation $\alpha_k$.

Suppose that, before the sequence, the permutation associated with the system is

$$\alpha_0 = \begin{pmatrix} 1 & 2 & \ldots & n \\ i_1 & i_2 & \ldots & i_n \end{pmatrix} \tag{40}$$



implying that the first event is the failure of the $i_1$-th fault. This event changes the magnitudes of all Coulomb stresses, so that the new state of the system is associated with a different permutation

$$\alpha_1 = \begin{pmatrix} 1 & 2 & \dots & n \\ j_1 & j_2 & \dots & j_n \end{pmatrix} \tag{41}$$

implying that the second event is the failure of the $j_1$-th fault, and so on. After the $(n-1)$-th event, the permutation is

$$\alpha_{n-1} = \begin{pmatrix} 1 & 2 & \dots & n \\ k_1 & k_2 & \dots & k_n \end{pmatrix} \tag{42}$$

implying that the last event is the failure of the $k_1$-th fault. Therefore the order of events in the sequence can be expressed as a permutation

$$\alpha^* = \begin{pmatrix} 1 & 2 & \dots & n \\ i_1 & j_1 & \dots & k_1 \end{pmatrix} \tag{43}$$

As to the order of events, the number of possible sequences in a system made of $n$ faults is equal to $n!$. Since every fault may slip only once in a sequence (assumption 4), there are $n!$ alternatives for the initial permutation $\alpha_0$, but only $(n-1)!$ for $\alpha_1$ and $(n-k)!$ for the generic permutation $\alpha_k$.

If the permutation after the $n$-th event is

$$\alpha_n = \begin{pmatrix} 1 & 2 & \dots & n \\ i & j & \dots & k \end{pmatrix} \tag{44}$$

the duration of the interseismic interval preceding the next sequence is

$$\Delta T = -\frac{x_i(t_n+)}{\dot{x}} \tag{45}$$

and the sequence will start with the failure of the $i$-th fault. In order to find out the relationship between the initial permutation $\alpha_0$ and the order of events given by $\alpha^*$, it is necessary to examine which are the orders of magnitude of the quantities controlling the evolution of the system.

## 6 Discussion

The evolution of the system is controlled by the stress rate $\dot{\sigma}$ and by the stress transfer matrix $\Delta x_{ij}$. We calculate the typical values of these quantities for seismic sequences.

For many seismogenic regions, typical strain rates are in the order of $10^{-15}$ to $10^{-14}$ s$^{-1}$. With $\mu = 30$ GPa and $\nu = 0.25$, Eq. (3) and Eq. (6) yield stress rates $|\dot{\sigma}|$ in the order of 2 kPa a$^{-1}$ for the lower value and of 20 kPa a$^{-1}$ for the higher value of strain rate. Calculation of $\Delta x_{ij}$ requires the knowledge of the seismic moments associated with each fault, of the fault areas and of the distances between them. A typical seismic moment of an event in the sequence can be calculated by assuming an



area $A_i = 100$ km$^2$ and an average slip $u_i = 0.5$ m, whence $m_i \simeq 10^{18}$ N m. With these values, Eq. (19) yields a stress drop $\Delta\sigma_i \simeq 1$ MPa. This value corresponds to the stress that is accumulated in a time interval $t_0 = 500$ a at a rate of 2 kPa a$^{-1}$ or $t_0 = 50$ a at a rate of 20 kPa a$^{-1}$. From Eq. (20) with a typical value $\epsilon = 0.7$ (Scholz, 1990), we obtain $\Delta\sigma_i = 0.6\,\tau_i$.

As to the distance between the centers of two neighbouring faults, we may roughly assume $r = 2\sqrt{A_i} = 20$ km. Then, according to Eq. (16) and Eq. (17), the stress $\Delta\sigma_{ij}$ transferred from a fault to its first neighbours is in the order of 10 kPa, so that the maximum value of $\Delta\sigma_{ij}$ ($i \neq j$) is in the order of one hundredth of stress drop. It must be noted that a greater value of $m_i$ does not entail a proportionally greater value of $\Delta\sigma_{ij}$, because it implies a greater fault area and a greater distance between faults.

An obvious effect of fault interaction is the shortening of time intervals between seismic events: for neighbouring faults, the gained time $\Delta\sigma_{ij}/|\dot\sigma|$ ranges from 5 to 0.5 a according to the value of $\dot\sigma$. Hence the maximum stress transferred by one event is equivalent to several months or several years of tectonic loading.

The magnitude of the differences $d_{ij}$ between Coulomb stresses is critical for the occurrence of a sequence. A lower limit for $d_{ij}$ is set by the magnitude of transferred stress $\Delta\sigma_{ij}$ ($i \neq j$). If $d_{ij}$ is smaller than $\Delta\sigma_{ij}$, the failure of the $i$-th fault would immediately produce the failure of the $j$-th fault, in contrast with assumption 5. Hence a condition for having a sequence of $n$ distinct events is $d_{ij} > \Delta\sigma_{ij}$ at any time.

An upper limit for $d_{ij}$ is set by the observed durations of seismic sequences. The $d_{ij}$ must be small enough that a sequence is completed within a few months, if we take into account the effect of stress transfer between faults. Hence we may assume as an upper limit for $d_{ij}$ the stress change $\dot x\,\delta t$ that tectonic loading produces in a time $\delta t \ll \Delta T$ (assumption 6) plus the sum of transferred stresses $\Delta x_{ij}$ ($i \neq j$). A greater value for $\delta t$ (several decades) can be assumed for lower stress rates, a smaller value (several years) for higher stress rates.

With this premise, we consider how the order $\alpha^*$ of events in a sequence is determined. The quantities determining $\alpha^*$ are the initial stress state of the fault system, the stress drops and the stress transfers associated with each event.

The simplest case is when the stress drops are about equal to each other and the $d_{ij}$ are always greater than the transferred stresses $\Delta x_{ij}$ ($i \neq j$). If these conditions are fulfilled, the only effect of the $k$-th event is to shift the label $i_k$ to the last position in the permutation $\alpha_k$, while the stress transfers $\Delta x_{i_k j}$ do not change the relative positions of the other labels. So we can associate with each event a permutation

$$\eta = \begin{pmatrix} i_1 & i_2 & \dots & i_n \\ i_2 & i_3 & \dots & i_1 \end{pmatrix} \tag{46}$$

such that

$$\alpha_k = \eta\,\alpha_{k-1} \tag{47}$$

It follows that the order of events is given by the initial permutation, i.e. $\alpha^* = \alpha_0$. The final permutation $\alpha_n$ is also equal to $\alpha_0$, but this does not imply the repetition of the order $\alpha^*$ in the following sequence. According to Eq. (36), the new sequence will start with different values of $d_{ij}$, that may produce a different order of events.





Apart from the case just described, the order implied by $\alpha_0$ is generally changed during a sequence, because the $d_{ij}$ have the same order of magnitude as the $\Delta x_{ij}$ $(i \neq j)$. In addition, if an event $j$ has a stress drop that is considerably greater than the others, the label $i_j$ will permanently occupy the last position in the permutation, thus altering the initial order. It follows that the order of events is different from the initial order of stresses, i.e. $\alpha^* \neq \alpha_0$. The final permutation $\alpha_n$ is also different from $\alpha_0$.

## 7  An example: the case $n = 3$

As an example, we consider a system made of three faults with strike-slip mechanism. We suppose that the faults are equal to each other, with distances $r_{12} = r_{23} = 20$ km between their centers. With a strain rate $\dot{e} = 10^{-14}$ s$^{-1}$, the stress rate calculated from Eq. (6) is $\dot{\sigma} \simeq 19$ kPa a$^{-1}$. For the sake of simplicity, we suppose that the faults have the same static friction $\tau = 1$ MPa and produce events with the same seismic moment $m_0 = 10^{18}$ Nm. The stress transfer matrix (15) is symmetric, with nondiagonal components $\Delta x_{12} \simeq 17$ kPa and $\Delta x_{13} \simeq 2$ kPa. With a typical value $\epsilon = 0.7$, Eq. (20) yields stress drops $\Delta x_i = 600$ kPa.

We consider coordinates $x_i/\tau$. The parallelepiped $S$ is a cube with unit edge, defined by the disequalities

$$-1 \leq x_i/\tau \leq 0 \tag{48}$$

and can be divided into 6 subsets $S_j$ corresponding to the 6 permutations of $N_3$. During the interseismic intervals, the representative point $\mathbf{x}(t)$ of the system draws an orbit parallel to the line

$$x_1 = x_2 = x_3 \tag{49}$$

An event occurs whenever the point reaches one of the coordinate planes.

As anticipated in section 3, we can rotate the coordinate system so that the $\xi_3$ axis coincides with line (49). It is easy to see that

$$\mathbf{R} = \begin{pmatrix} b & -c & -a \\ -c & b & -a \\ a & a & a \end{pmatrix} \tag{50}$$

where

$$a = \frac{1}{\sqrt{3}}, \quad b = \frac{1+a}{2}, \quad c = \frac{1-a}{2} \tag{51}$$

The projection of $S$ on the plane $\xi_3 = 0$ is a regular hexagon $H$ with side $a$ (Fig. 2a). It is divided into 6 equilateral triangles $H_k$, that are the projections of the subsets $S_k$. We can follow the evolution of the system by looking at the projection $P$ of the representative point on $H$. During the interseismic intervals, $P$ does not change, because the representative point moves on a line perpendicular to $H$, and is close to the origin, because the $d_{ij}$ are much smaller than $\tau$. Whenever an event takes place, $P$ moves to a different subset of $H$.



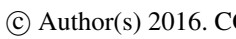

Suppose that, at a certain instant $t_0$ of the interseismic interval preceding a sequence, the state of the system is

$$\mathbf{x}_0 = \mathbf{x}(t_0) = (x_{01}, x_{02}, x_{03}) \tag{52}$$

with $x_{01} > x_{02} > x_{03}$. The associated permutation is then

$$\alpha_0 = \begin{pmatrix} 1 & 2 & 3 \\ 1 & 2 & 3 \end{pmatrix} \tag{53}$$

and the representative point $P_0$ belongs to the subset of $H$ labeled with 123 in Fig. 2b. We choose the vector $\mathbf{x}_0$ in order that the sequence is made of 3 distinct events, occurring in the order given by $\alpha_0$. Then $d_{12}$ and $d_{13}$ must be positive, with $d_{12} < d_{13}$. According to the discussion in the previous section,

$$\Delta x_{12} < d_{12} < d_{13} < \Delta x_{13} + \Delta x_{23} + \dot{\sigma} \delta t \tag{54}$$

where we assume $\delta t = 1$ a, so that $\dot{\sigma} \delta t \simeq 19$ kPa. Then $17\,\mathrm{kPa} < d_{12} < d_{13} < 38\,\mathrm{kPa}$ and we choose $d_{12} = 20$ kPa and $d_{13} =$
25 kPa. At the beginning of the sequence, the state is

$$\mathbf{x}(t_1-) = (0, d_{21}, d_{31}) \tag{55}$$

The mean is $\bar{x} = -15$ kPa, with a standard deviation $s = 10$ kPa. The state immediately after the first event is

$$\mathbf{x}(t_1+) = (\Delta x_{11}, d_{21} + \Delta x_{12}, d_{31} + \Delta x_{13}) \tag{56}$$

According to Eq. (47), the associated permutation is $\alpha_1 = \eta \alpha_0$ with

$$\eta = \begin{pmatrix} 1 & 2 & 3 \\ 2 & 3 & 1 \end{pmatrix} \tag{57}$$

and the representative point $P_1$ belongs to the subset of $H$ labeled with 231 in Fig. 2b. From Eq. (29), the time interval between the first and the second event is $\Delta t_1 = 66$ d. At the end of this interval, the state is

$$\mathbf{x}(t_2-) = (x_1(t_1+) + \dot{\sigma} \Delta t_1, 0, x_3(t_1+) + \dot{\sigma} \Delta t_1) \tag{58}$$

and the second event takes place. The state becomes

$$\mathbf{x}(t_2+) = (x_1(t_2-) + \Delta x_{21}, \Delta x_{22}, x_3(t_2-) + \Delta x_{23}) \tag{59}$$

The associated permutation is $\alpha_2 = \eta \alpha_1$ and the representative point $P_2$ belongs to the subset of $H$ labeled with 312 in Fig. 2b. The time interval between the second and the third event is $\Delta t_2 = 56$ d. At $t = t_3-$ the state is

$$\mathbf{x}(t_3-) = (x_1(t_2+) + \dot{\sigma} \Delta t_2, x_2(t_2+) + \dot{\sigma} \Delta t_2, 0) \tag{60}$$





The third event takes place and the state becomes

$$\mathbf{x}(t_3+) = (x_1(t_3-) + \Delta x_{31}, x_2(t_3-) + \Delta x_{32}, \Delta x_{33}) \tag{61}$$

with $\alpha_3 = \eta\alpha_2$, that coincides with $\alpha_0$. The representative point $P_3$ belongs to the subset of $H$ labeled with 123 in Fig. 2c. According to Eq. (26), the duration of the sequence is $\Delta t = 122$ d. In the absence of interaction, the duration would have been

$\Delta t' = 482$ d from Eq. (28). The evolution of the three components in time is shown in Fig. 3.

The final state is very different from the initial one, with a value of $\bar{x}$ that is about 40 times greater, while $s$ has not changed. The differences between components are $d_{12} = 5$ kPa and $d_{13} = 25$ kPa, so that $x_2$ has become closer to $x_1$ and farther from $x_3$. In fact, the state no longer fulfills the condition $d_{12} > \Delta x_{12}$ entailing that, when the next sequence occurs, the stress $\Delta x_{12}$ transferred to fault 2 by the slip of fault 1 will induce the immediate failure of fault 2.

According to Eq. (45), the duration of the time interval before the next sequence is $\Delta T \simeq 30$ a. Even though the order of events in this sequence is the same as in the previous one, the durations of the intervals $\Delta t_k$ are different. Moreover the stress distribution is altered in such a way that $\alpha_3 \neq \alpha_0$, entailing that even the order of events will be different in the following sequence.

The example shows that, even in a simple case as the one considered, the changes in $d_{ij}$ occurring in a sequence inevitably

change the characteristics of the following sequences. A further source of change intervenes when the events have different sizes, as shown in the following section.

## 8 The 2012 Emilia sequence

We consider the 2012 Emilia (Italy) seismic sequence, that was made of seven events with magnitudes between 5 and 6 (Pezzo et al., 2013). They occurred in the period between May 20th and June 3rd, 2012, and can be ascribed to a fault system

approximately lined up in the west-east direction, with a total length of about 50 km. The faults are all of thrust type and with shallow hypocenters between 5 and 10 km of depth.

Our aim is not to simulate this sequence in detail, but to use it as an example of a complex sequence for which the present model can afford the retrieval of the initial and final stress states. According to the model, we approximate the real fault system with a set of $n = 7$ faults having the same strike and dip angles and the same average depth. If we number the faults from west

to east, the order of events is given by

$$\alpha^* = \begin{pmatrix} 1 & 2 & 3 & 4 & 5 & 6 & 7 \\ 5 & 6 & 7 & 4 & 3 & 2 & 1 \end{pmatrix} \tag{62}$$

Therefore, fault slip started about in the middle of the system and propagated eastward up to the end of the system (5, 6, 7); then it propagated from the middle to the west end (4, 3, 2, 1).

The data required for retrieving the initial and final states according to Eq. (31) and Eq. (32) are the elastic and frictional

properties of the medium (the rigidity $\mu$, the Poisson ratio $\nu$, the coefficient $\kappa$ of static friction), the geometry of the faults (the areas $A_i$, the distances $r_{ij}$, the dip angle $\delta$), the strain rate $\dot{e}$, the occurrence times $t_i$ and the seismic moments $m_i$ of the events.





As to the elastic and frictional properties, we take $\mu = 30$ GPa, $\nu = 0.25$ and an effective coefficient of friction $\kappa = 0.6$. The areas and the locations of the faults have been inferred by employing the distances between hypocenters along the strike direction as constraints (Caporali and Ostini, 2012; Serpelloni et al., 2012). The distances between the centres of the faults are $r_{12} = r_{23} = r_{67} = 5$ km, $r_{34} = r_{56} = 8$ km, $r_{45} = 12$ km. The projection of the faults on a vertical plane is shown in Fig. 4.

We treat all sources as pure reverse, dip-slip faults with $\delta = 40°$, an average of the values given by Convertito et al. (2013). The strain rate is $\dot{e} = -3 \times 10^{-15}$ s$^{-1}$ (Caporali and Ostini, 2012). The moments $m_i$ are derived from the moment magnitudes reported in Tramelli et al. (2014), whereas fault slips $u_i$ are calculated from $m_i$ and $A_i$. The data are shown in Table 1.

From these data we calculate the rate $\dot{x}$ of Coulomb stress and the stress transfer matrix $\Delta x_{ij}$. From Eq. (10), $\dot{x} \simeq 2$ kPa a$^{-1}$. The initial state $\mathbf{x}(t_1-)$ and the final state $\mathbf{x}(t_7+)$ are shown in Fig. 5. The initial and final permutations are respectively

$$\alpha_0 = \begin{pmatrix} 1 & 2 & 3 & 4 & 5 & 6 & 7 \\ 5 & 4 & 7 & 1 & 2 & 3 & 6 \end{pmatrix} \tag{63}$$

$$\alpha_7 = \begin{pmatrix} 1 & 2 & 3 & 4 & 5 & 6 & 7 \\ 6 & 7 & 1 & 2 & 4 & 3 & 5 \end{pmatrix} \tag{64}$$

The evolution of the system during the sequence shows that Eq. (47) does not hold for any value of $k$. Therefore $\alpha_7$ is different from $\alpha_0$ and they are both different from $\alpha^*$. This is a consequence of the heterogeneous distribution of seismic moment in the fault system. In particular, the evolution of stress was conditioned by the first and the fourth event, due to the failures of faults 5 and 4 respectively, having greater seismic moments than the average. As a consequence of the greater stress drop, fault 5 permanently occupies the last position in all permutations from $\alpha_1$ to $\alpha_7$. The stress transfers associated with each event also play a role in determining the evolution of the sequence, contributing to the rearrangement in the permutations. Due to the many stress transfers to fault 6, Eq. (28) yields $\Delta t' \simeq 46$ a for the duration of the sequence in the absence of fault interaction, a much longer time than the observed duration $\Delta t \simeq 15$ d.

Figure 6 shows that, at the beginning of the sequence, the mean Coulomb stress was $\bar{x} \simeq -0.05$ MPa, with a standard deviation $s \simeq 0.03$ MPa, whereas at the end of the sequence $\bar{x} \simeq -1.2$ MPa and $s \simeq 0.4$ MPa. Therefore Coulomb stresses are much more spread out after the sequence than before, since the standard deviation is one order of magnitude larger.

According to Eq. (64), the faults with the highest values of $x_i$ after the sequence are the 6th, 7th and 1st, showing that, in the absence of external perturbations, the next sequence would start in the proximity of one end of the system, rather than in the middle as the 2012 sequence. According to Eq. (45), the next sequence will take place after an interseismic interval $\Delta T \simeq 440$ a. This figure appears to be representative of typical recurrence times of moderate-size earthquakes in this area: the largest event before the 2012 sequence was the $M_w$ 5.5 November 17, 1570, Ferrara earthquake (Rovida et al., 2011).

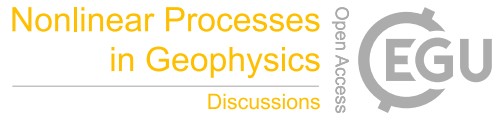

# 9 Conclusions

The aim of this study was to enlighten the conditions allowing the occurrence of seismic sequences and the processes controlling the order of events in a sequence. We considered a model of a fault system made of $n$ coplanar faults subject to a constant strain rate. The state of the system has been described by the $n$ values of the Coulomb stresses of the faults.

At any instant of time, the order of Coulomb stresses according to their magnitudes can be expressed as a permutation of the first $n$ integers. Therefore a permutation can be associated with each state of the system and the state space can be divided into $n!$ subsets corresponding to the $n!$ permutations. The permutation associated with the system does not change as long as the system is at rest, so that the state remains in the same subset. Whenever a seismic event takes place, the order of Coulomb stresses is changed and is expressed by a different permutation: an event corresponds to a switch of the state vector to a different

subset of the state space.

We are now in a position to answer the questions we asked in the Introduction.

1) As to the stress conditions allowing the occurrence of seismic sequences, a crucial role is played by the differences between the Coulomb stresses of faults. A lower limit for these differences is set by the magnitude of transferred stresses and an upper limit by the observed durations of seismic sequences. This constrains their values in a narrow interval in the order of

tens of kPa.

2) The order of events in a sequence is determined by the initial distribution of Coulomb stresses, by the stress drops and by the stress transfers associated with each event. The order of events that is implicit in the initial state is generally modified by the changes in the state vector intervening during the sequence. The dominant contribution to stress changes is given by stress drops, which are typically hundred times greater than stress transfers. However stress transfers have a major role both in

anticipating the occurrence times of the events and in altering the order of events implicit in the initial state.

3) The model shows that the characteristics of consecutive sequences originated by a fault system are bound to change. The order implicit in the initial stress distribution is generally changed during the sequence, because the stress transfers between faults have the same order of magnitude as the differences in Coulomb stresses and because one or more events in the sequence may have greater stress drops than the others. In all cases, the state of the system at the end of a sequence is different from

the initial one, entailing that the durations of the interseismic intervals between consecutive sequences and between events in a sequence are different.

4) On the basis of the model, if we observe a seismic sequence, we can retrieve the state of stress at any time during the sequence. In particular, we can calculate the state of the system at the beginning and at the end of the sequence. This has been done as an application for the 2012 Emilia (Italy) seismic sequence, which was made of seven events with magnitudes between

5 and 6. In this case, the evolution of stress was conditioned by the first and the fourth event, having greater seismic moments than the average. The model shows the complex interplay between fault dislocations that produced the observed order of events, resulting in a greater stress heterogeneity at the end of the sequence. It predicts that, in the absence of external perturbations, the next sequence will occur after an interseismic interval of a few centuries and will be completely different from the 2012 sequence.



## Appendix A

We consider two different sources: a point-like dislocation (a double couple of forces) in an unbounded elastic medium and a finite square dislocation in an elastic half-space. For both sources, we calculate the tangential traction $\sigma_t$ produced on the fault plane in the slip direction, as a function of the distance from the source along the strike direction $x$.

We assume that the elastic medium is a Poisson solid with rigidity $\mu$. The fault lays on the plane $y = 0$ and its center is in the origin of the coordinate system. Let $n_i$ be the unit vector perpendicular to the fault and $m_i$ be the unit vector in the slip direction, so that

$$n_i = (0, 1, 0), \quad m_i = (\cos\theta, 0, \sin\theta), \tag{A1}$$

where $\theta$ is the rake angle. The tangential traction in the slip direction is then

$$\sigma_t = \sigma_{ij} m_i n_j, \tag{A2}$$

where $\sigma_{ij}$ is the stress tensor. We compare the tractions produced by the two sources in two cases: a strike-slip fault ($\theta = 0$) and a dip-slip fault ($\theta = \pi/2$). In the two cases, we have $\sigma_t = \sigma_{xy}$ and $\sigma_t = \sigma_{yz}$ respectively. Let $m_0$ be the scalar seismic moment of the dislocation.

a) In the case of a point-like source, the moment tensor of the equivalent double couple has nonvanishing components

$$M_{xy} = -m_0 \cos\theta, \quad M_{yz} = -m_0 \sin\theta. \tag{A3}$$

The displacement is

$$u_i = M_{jk} G_{ij,k}, \tag{A4}$$

where

$$G_{ij} = \frac{1}{8\pi\mu} \left( r_{,kk}\, \delta_{ij} - \frac{2}{3} r_{,ij} \right) \tag{A5}$$

is the Somigliana tensor and

$$r = \sqrt{x^2 + y^2 + z^2}. \tag{A6}$$

In the case $\theta = 0$, we have

$$\sigma_t = \mu(u_{x,y} + u_{y,x}). \tag{A7}$$

Setting $y = 0$ and $z = 0$, we obtain

$$\sigma_t(r) = \frac{5m_0}{12\pi r^3}, \tag{A8}$$



where $r = |x|$. In the case $\theta = \pi/2$, we have

$$\sigma_t = \mu(u_{y,z} + u_{z,y}). \tag{A9}$$

Setting again $y = 0$ and $z = 0$, we obtain

$$\sigma_t(r) = \frac{m_0}{6\pi r^3}. \tag{A10}$$

b) The displacement produced by a finite, rectangular source in an elastic half-space was given by Okada (1992). We consider a square source with side $L$ and center at depth $L$. The dip angle is $\delta = \pi/4$. The analytical expressions of $\sigma_t(r)$ are too complicated to be reported here and we only show their graphs.

A comparison between the two solutions is shown in Fig. A1.

**Appendix B**

Solutions for a point-like dislocation in a homogeneous and isotropic poroelastic medium are given by Cheng and Detournay (1998) and Carvalho and Curran (1998). The stress field is made of a constant term (the coseismic stress) plus a time-dependent term associated with fluid diffusion. We introduce a dimensionless variable

$$\xi(t) = \frac{r_{ij}}{2\sqrt{ct}} \tag{B1}$$

where $c$ is the hydraulic diffusivity and a coefficient

$$a = \frac{\nu_u - \nu}{(1 - \nu)(1 - \nu_u)} \tag{B2}$$

where $\nu_u$ is the undrained Poisson modulus. When the $i$-th fault slips with moment $m_i$, the additional tangential traction on the $j$-th fault is

$$\Delta\sigma'_{ij}(t) = \frac{am_i}{2\pi r_{ij}^3} f(t) \tag{B3}$$

where

$$f(t) = -\frac{2}{\sqrt{\pi}} \xi e^{-\xi^2} + 3\frac{\operatorname{erf}\xi}{\xi^2} - \frac{6}{\sqrt{\pi}}\frac{e^{-\xi^2}}{\xi} + \operatorname{erfc}\xi \tag{B4}$$

for strike-slip and

$$f(t) = -\frac{3}{4}\frac{\operatorname{erf}\xi}{\xi^2} + \frac{3}{2\sqrt{\pi}}\frac{e^{-\xi^2}}{\xi} - \frac{1}{2}\operatorname{erfc}\xi \tag{B5}$$

for dip-slip (Fig. B1). For $t \to \infty$, the traction (B3) approaches an asymptotic value

$$\Delta\sigma_{ij}^\infty = \frac{am_i}{2\pi r_{ij}^3} \tag{B6}$$





for strike-slip and

$$\Delta\sigma_{ij}^{\infty} = -\frac{am_i}{4\pi r_{ij}^3} \tag{B7}$$

for dip-slip. According to the choice of a Poisson solid, we take $\nu_u = 0.25$. For a typical value $\nu = 0.2$ under drained conditions (e.g. Rice and Cleary, 1976), it results $a \simeq 0.1$. Then the ratio $|\Delta\sigma_{ij}^{\infty}|/\Delta\sigma_{ij}$ between the asymptotic poroelastic stress and the coseismic stress is about 0.12 for strike-slip and 0.15 for dip-slip. These are the maximum values, that may be reached for $t \gg \tau$, where $\tau = r_{ij}^2/(4c)$ is the characteristic diffusion time. For distances of tens of km, $\tau$ is much longer than the typical duration of a seismic sequence, so that the poroelastic effect is at least one order of magnitude smaller than the coseismic stress transfer.

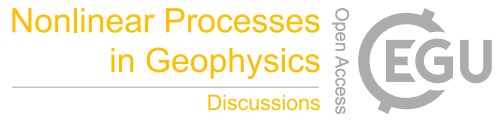

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

## Figure Captions

Fig. 1 - Sketch of the model with $n$ coplanar faults.

Fig. 2 - Case $n = 3$: (a) The hexagon $H$ with its subsets labeled by the associated permutation; (b) States of the system during a seismic sequence: each point belongs to the subset labeled by the associated permutation; (c) magnification of $H$ showing the initial and final states.

Fig. 3 - Case $n = 3$: components of the state vector $\mathbf{x}$ as functions of time during the seismic sequence shown in Fig. 2.

Fig. 4 - Geometry of the model for the 2012 Emilia seismic sequence. Stars indicate the hypocenters.

Fig. 5 - Components of the state vector $\mathbf{x}$ at the beginning (a) and at the end (b) of the 2012 Emilia seismic sequence, as calculated from the model. The mean $\bar{x}$ and the standard deviation $s$ are shown.

Fig. A1 - Tangential traction produced by a dislocation in the slip direction on the fault plane in the case of a point-like source in an unbounded medium (solid lines) and a finite source in a half-space (dashed lines): (a) strike-slip; (b) dip-slip.

Fig. B1 - Function $f(t)$ in the case of strike-slip (solid line) and dip-slip (dashed line). Time is in units of the characteristic diffusion time $\tau$.

**Table 1.** Data for the seismic events of the 2012 Emilia sequence. See Fig. 4 for fault numbers.

| Event | Fault | $t_i\,(\mathrm{d})$ | $m_i\,(\mathrm{N\,m})$ | $A_i\,(\mathrm{km}^2)$ | $u_i\,(\mathrm{m})$ |
|-------|-------|------|----------|---------|--------|
| 1 | 5 | 0 | $8.9\cdot10^{17}$ | 60 | 0.49 |
| 2 | 6 | 0.0025 | $5.6\cdot10^{16}$ | 16 | 0.12 |
| 3 | 7 | 0.47 | $5.6\cdot10^{16}$ | 16 | 0.12 |
| 4 | 4 | 9.2 | $6.3\cdot10^{17}$ | 60 | 0.35 |
| 5 | 3 | 9.4 | $1.1\cdot10^{17}$ | 16 | 0.23 |
| 6 | 2 | 9.4 | $7.9\cdot10^{16}$ | 16 | 0.16 |
| 7 | 1 | 15 | $5.6\cdot10^{16}$ | 16 | 0.12 |

**Figure 1.**

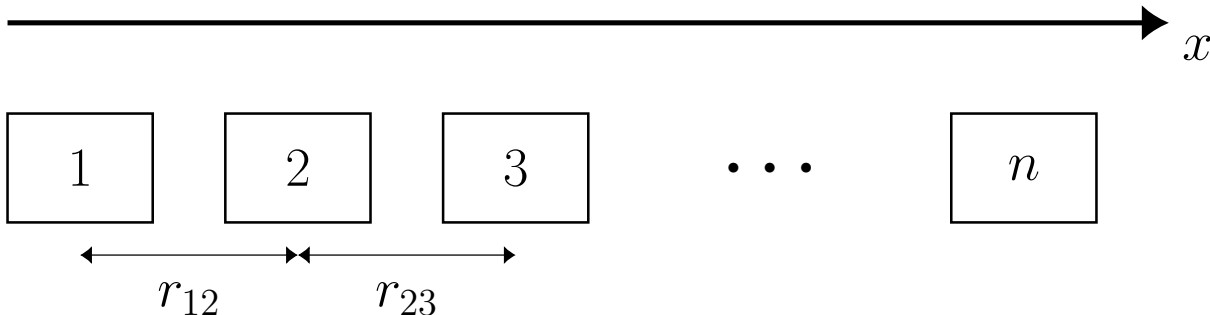





**Figure 2.**

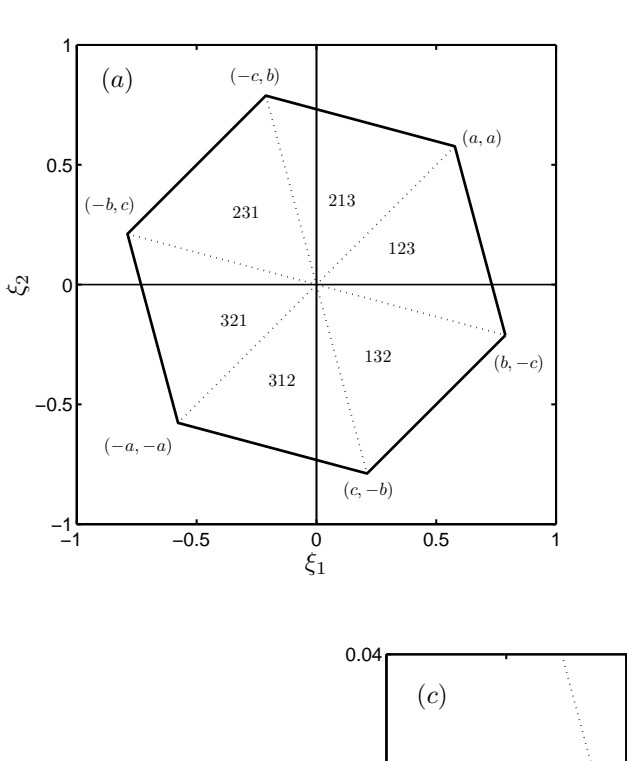

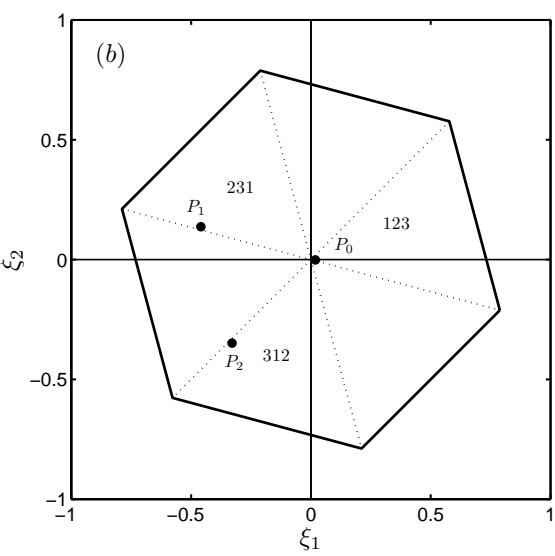

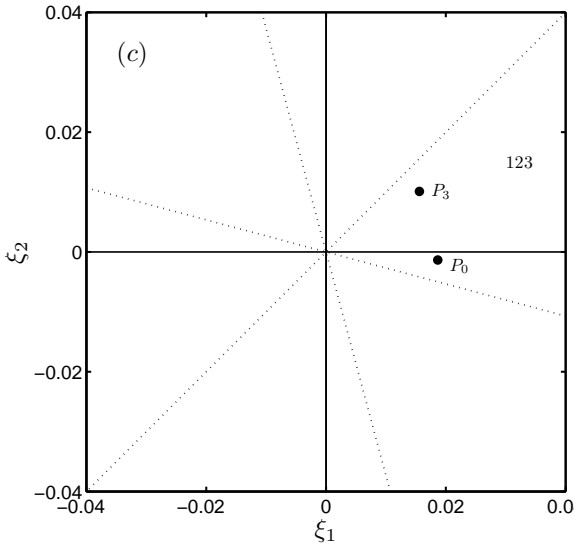





**Figure 3.**

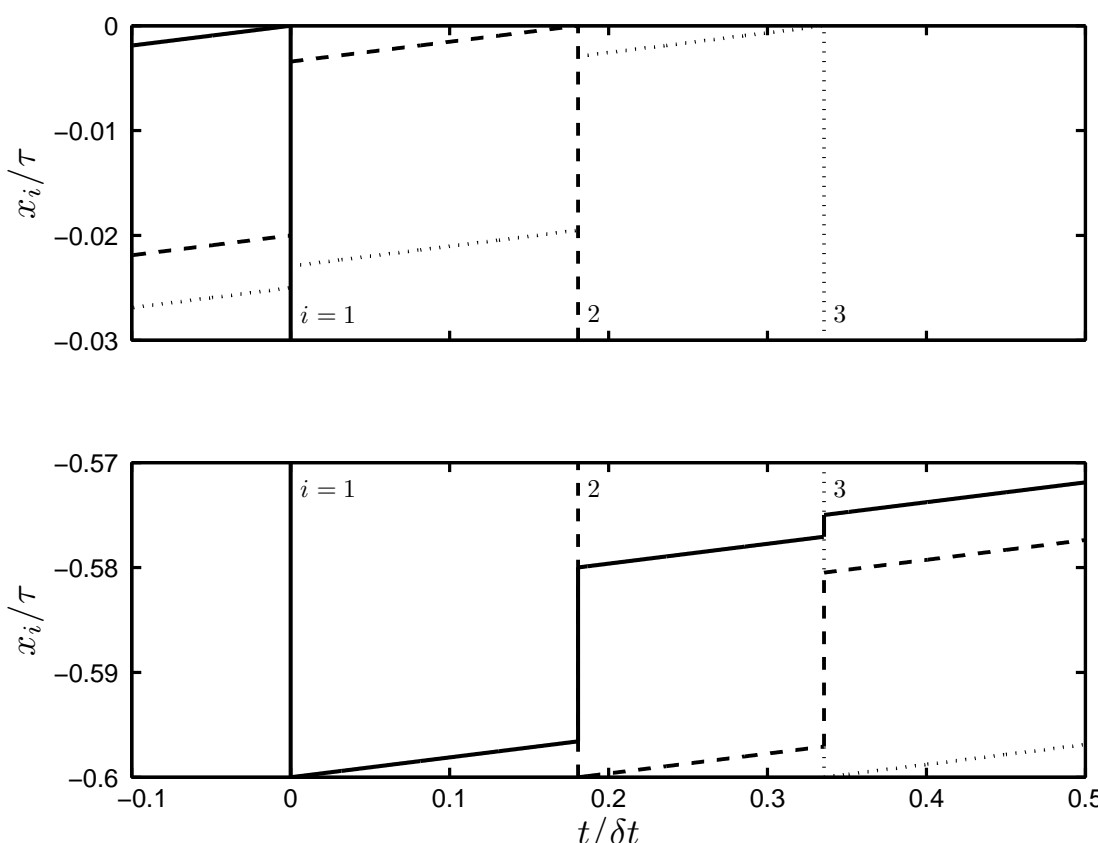

**Figure 4.**

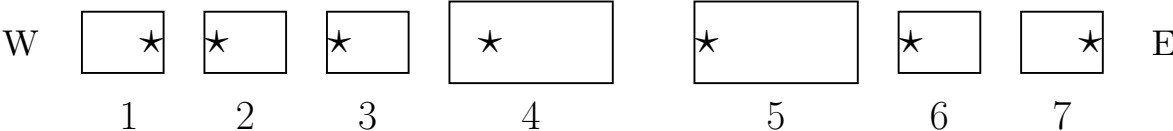



**Figure 5.**

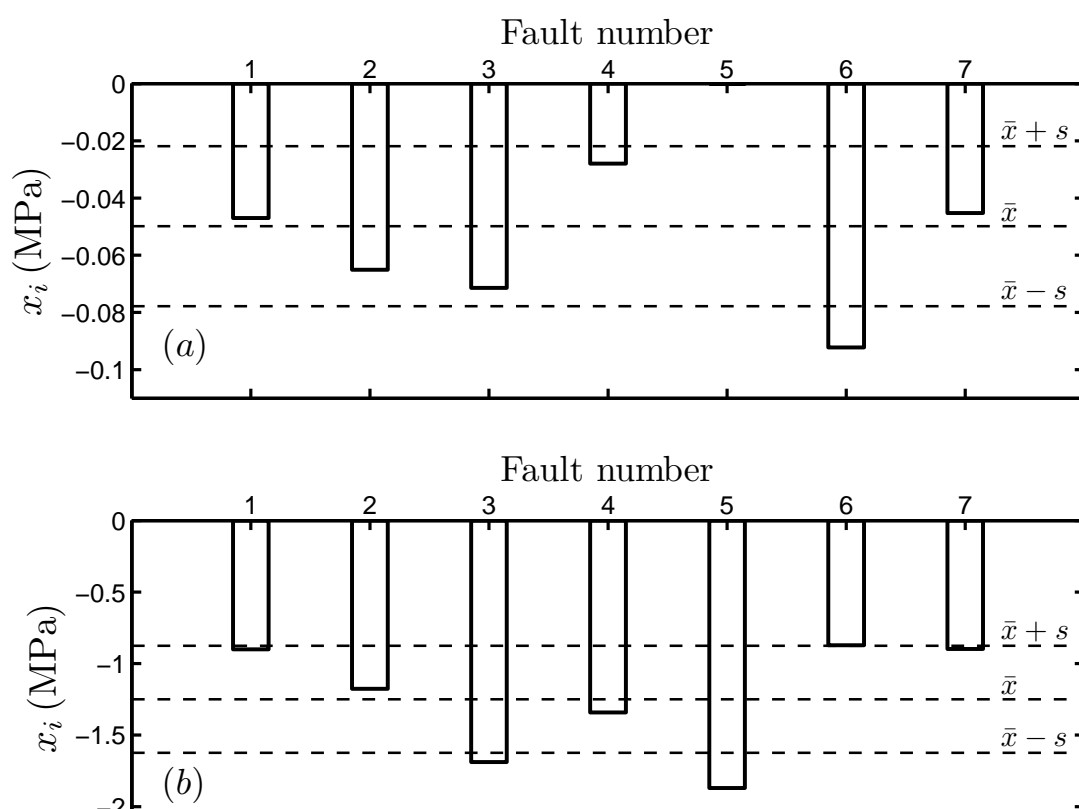

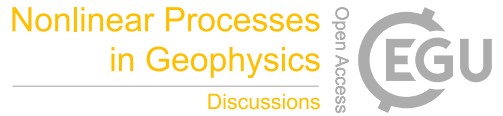



**Figure A1.**

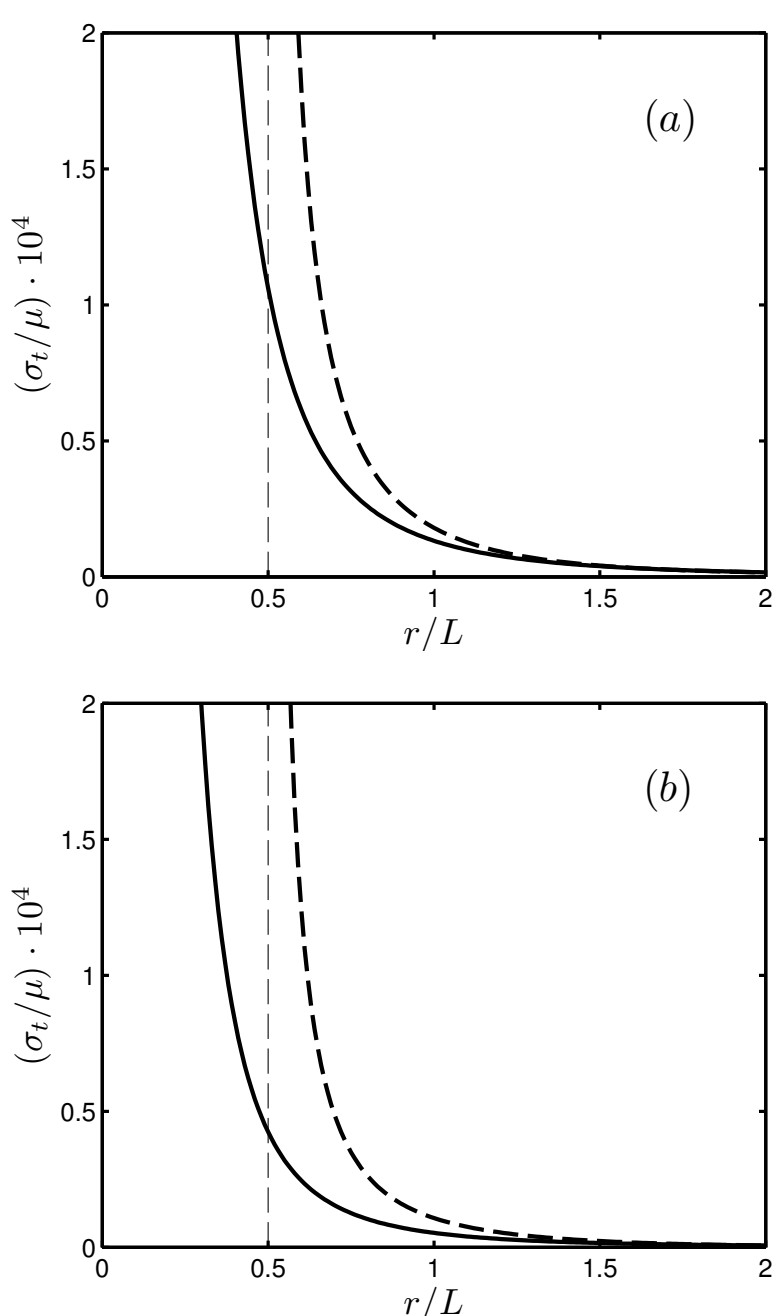





**Figure B1.**

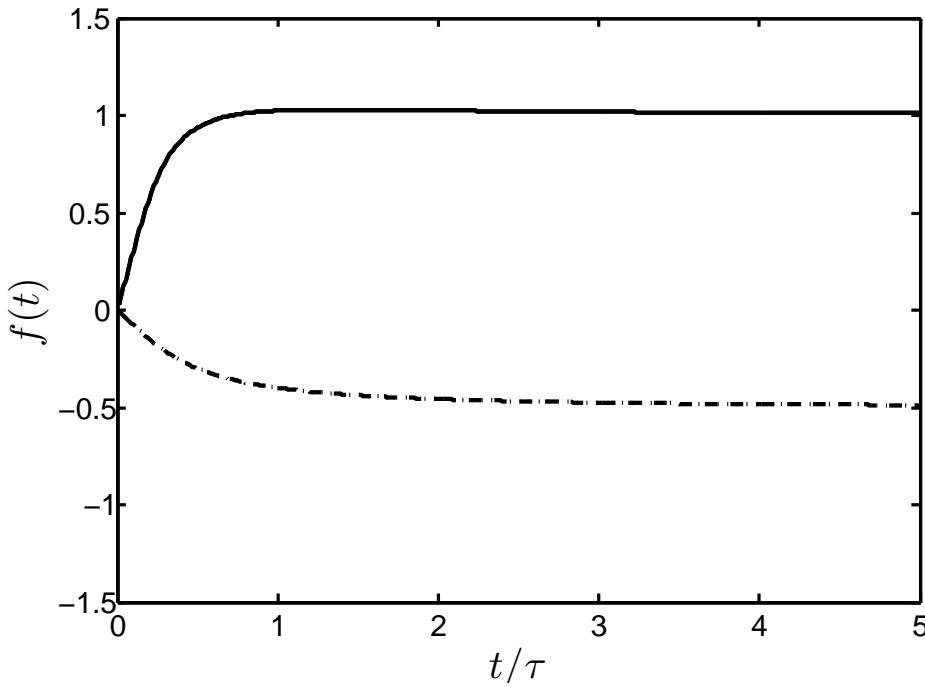