# Peer review of "Conditions for the occurrence of seismic sequences in a fault system"

_Nonlinear Processes in Geophysics, 2016_

## Referee Comment (RC1) · Anonymous Referee #1 · 4 Aug 2016

This work is interesting and potentially significant; however, that significance is not clear in the current version.

More details on the data and seismic sequence are required, particularly the 2012 Emilia sequence. This should include details on data source, plots of historic time series and, again, the Emilia sequence in particular. I suggest a map showing the location of the events and the associated fault network, where available.

Seven is a lot of assumptions or constraints. In particular, assumption 6 concerns me. How do we know that the sequence time is short versus interevent time? Again, more details on the sequence time series would help to inform that question. It also is necessary to expand on all assumptions, particularly 4, 5 and 6.

The abstract is unclear and does not preface the work properly - it favors details without sufficient motivation or background to place the research in context, particularly for the uninitiated.

My biggest reservation is the significance of the work and what is innovative or new here. Based on the substantial body of work in this area, the results here are not particularly surprising - what is new and significant in this work is not clear from the results or conclusions. We would expect, for example, for the sequence to change in time as a result of the Coulomb interactions. Part of the problem is the large volume of research in this area. The original work of Stein and King has expanded (Toda, Jaume, Sykes and others) and evolved to study many different cases. In particular, many researchers have studied CFF interactions in network fault systems, many that include realistic geometry and often are more complicated. For example, Rundle and others have studied Virtual California for many years (references 1988 through the present), a simplified CA fault model that produces complex seismic time histories based on CFF interactions, Steacey, referenced in this paper, is only one of a number of researchers that have looked at Coulomb stress interactions in detail. Others have added other features and phenomena (Ben-Zion, Dieterich, Main, among others). Simple models with CFF are less common but there is still a substantial body of work, including CA models. Placing this research in context would help, but is not sufficient. The authors need to revise the text, particularly the results and conclusions, to make it clear what is new and innovative and/or interesting and significant about this research.

Minor revisions: The paper is missing many references. For example, on p. 2 (∼line 5), the authors state 'Fault interaction and its role in earthquake triggering have been widely studied (e.g. Steacy et al., 2005).' Fault interaction and triggering have been researched for decades and a large volume of literature accompanies that work. Although it is not appropriate to list them all, the salient work should be cited. Or, on p. 4, the double-couple point source is discussed and should be referenced. The manuscript should be rewritten with this in mind.

The figures are not detailed well enough, and their motivation or use is not clear; this applies, in particular, to Figure 2.

---

## Referee Comment (RC2) · Anonymous Referee #2 · 29 Aug 2016

Reviewer's comments

"Conditions for the occurrence of seismic sequences in a fault system" by Michele Dragoni and Emanuele Lorenzano submitted to Nonlin. Processes in Geophysics (2016)

Summary: Dragoni and Lorenzano present textbook-style arguments favouring conditions under which a pre-existing fault system could generate a seismic sequence. Seismic sequences, as observed in 1997 Umbria-Marche sequence and 2012 Emilia sequence, have been subjects of detailed studies. In this regard, the use of the elastic model for seismic sequencing and the changes in the differences between Coulomb stresses and different stress drops of the events resulting in alteration of the initial

permutation order during sequencing are important points. Furthermore, for seismic sequences similar to the Emilia 2012 sequence, the authors claim that the state of stress at any time during the sequence can be retrieved.

The model used by the authors makes a few assumptions. With the exception of the external perturbation not influencing the system of n coplanar faults, the remaining assumptions are acceptable. Clearly, the authors exercise caution about the non-use of external perturbation in making long-term predictions.

General:

(1) The arguments in favour of the permutation order of the sequence are clear. However, they are uniquely based on static Coulomb stress field changes. These are in contrast to the arguments presented by Convertito, Catalli, and Emolo (Scientific Reports, 2013, DOI: 10.1038/srep03114). Their main conclusion is that static stress distribution alone does not explain the location of the subsequent events in the seismic sequence. Furthermore, their argument favouring dynamic triggering to influence the seismic sequence would have to be looked at carefully.

(2) Dragoni and Lorenzano argue that the pore-fluid effects are one order of magnitude smaller than the Coulomb stress changes. This is in contrast to the argument by Convertito, Catalli, and Emolo (2013) on the variation in permeability and pore-pressure effects due to a massive presence of fluids in the Po Plain basin to play a triggering role in the seismic sequence.

(3) Source mechanism of major events in observed seismic sequences in northern Italy or central Italy falls either into a reverse-fault or normal fault mechanisms. The seismic sequence studied in the 2012 Emilia region satisfies the conditions proposed for similar fault systems. Castro et al. (Geophysical Journal International, 2013) suggest that the reverse faulting events of the Emilia 2012 sequence generated low stress drops but relatively large amounts of low frequency effects. This is an important point to be considered.

(4) The Umbria-Marche sequence and the Amatrice-Composto-L'Aquilla sequences reveal locations in the vicinity of original fault systems in central Italy. This suggests that the present static model might require revisiting for forecasting purposes.

Specific:

(1) References: The list of references is adequate but a brief review of the findings of other authors on the 2012 Emilio sequence is missing in the text. (2) Table: References should be included citing where the data information came from. (3) Figures: Adequate for the purpose of showing how the static model works for the sequence.
* * *

---

## Author Comment (AC1) · 27 Sep 2016

**Response to Reviewer 1**

1) The reviewer suggests that we give more details on the 2012 Emilia sequence.

We agree to his request and shall include more details on the data and on their sources in Table 1 and two new figures (numbered as 4 and 5 in the revised version):

a map showing the location of the events and a plot of their seismic moments.

2) The reviewer also suggests that we expand the discussion of the model assumptions, in particular assumptions 4, 5 and 6.

These assumptions are suggested by the features of the sequences we are describing, as recorded in seismic catalogues (e.g. Rovida et al., 2011). Assumptions 4 and 5 follow from the observation that sequences are made of distinct events, each one associated with the failure of a distinct fault in the system, and there is no reactivation of the same fault during a sequence. Assumption 6, stating that the sequence duration is short versus the intersequence time, is justified by seismic history, showing that intersequence intervals are in the order of centuries, while sequence durations are in the order of weeks or months. These comments will be added in section 2.

3) The reviewer finds that the abstract is unclear and favors technical details rather than focusing on the general conclusions of the paper.

The abstract will be rewritten and we shall stress that the key point of the model is to show that the knowledge of the order of activation of faults in a seismic sequence yields information on the state of the fault system before and after the sequence. The concept of permutation is crucial to this aim, because the evolution of the system can be expressed as a sequence of permutations and the order of activation can be described by a particular permutation of the n faults.

4) The reviewer suggests revising the text, particularly the results and conclusions, to make it clear what is new and innovative and/or interesting and significant about this research.

[Figure]

As stated in the Introduction, the aim of the paper is to answer some basic questions concerning seismic sequences. When we observe a sequence, we acknowledge that it is due to a system of n faults that fail one after the other. However, we do not know why the faults fail in that particular order. The order must be a consequence of the initial stress state of the fault system and of the mutual interaction between the faults of the system during the sequence. We show that the knowledge of the order of activation of faults in the sequence yields information on the state of the fault system before and after the sequence. To this aim, we introduce the concept of permutation of the n faults, ranking the faults according to the magnitudes of their Coulomb stresses. Such a permutation describes the state of the system at a given time and changes whenever a fault is activated. The order of activation itself can be described by a particular permutation of the faults. These considerations will be added in the Conclusions of the paper.

5) The reviewer notices that the paper is missing references concerning the subject of fault interaction and triggering.

Following this suggestion, we shall add references to Stein et al. (1992), Harris (1998), Stein (1999), Gomberg et al. (2000), Belardinelli et al. (2003). We shall also add a reference (Love, 1927) for the double-couple point source solution.

6) The reviewer considers that the figures are not detailed well enough, and their motivation or use is not clear: this applies, in particular, to Figure 2.

Figure 2 is a graphical illustration of the evolution of a system made of three faults (n = 3). This case is considered because it can be illustrated graphically, owing to the

small number of variables involved. Cases with n > 3 would require higher dimensional spaces. The graphical representation allows a better understanding of the evolution of the state of the fault system during a seismic sequence. This explanation will be added at the beginning of section 7. The captions of figures 2 and 3 will be rewritten and more details will be added in the other captions.

**Response to Reviewer 2**

1-2) The reviewer suggests a discussion of some results obtained by other authors with regard to the 2012 Emilia sequence.

We agree with his request, though the aim of the present work is not to reproduce the details of any particular seismic sequence, but to show how the knowledge of the activation order of faults can give information on the stress state of a fault system. Convertito et al. (2013) suggest that dynamic triggering may have had a role in influencing the seismic sequence, in addition to the variation in permeability and pore-pressure effects due to a massive presence of fluids in the Po Plain basin. We neglected the effect of pore fluid diffusion, on the basis of general considerations in Appendix B, and did not consider dynamic triggering. If these effects are relevant and are introduced in the calculations, they may alter the sequence of permutations and yield a final permutation different from (64). However, they will not change the general conclusions of the paper. A mention of the possible role of pore fluids will be added in section 2, when the argument is introduced. A short discussion will be added in section 8 and will be recalled at the end of the Conclusions.

3) The reviewer mentions results obtained by Castro et al. (2013) from seismic spectra, suggesting that the events of the Emilia sequence generated relatively low stress drops.

We shall mention this paper in section 8 and show that our model yields values ranging between 0.9 and 1.9 MPa, within the range obtained in that study.

4) As suggested by the reviewer, references will be included in Table 1.

**References**

Belardinelli, M. E., Bizzarri, A. and Cocco, M.: Earthquake triggering by static and dynamic stress changes, J. Geophys. Res., 108 (B3), 2135, doi: 10.1029/2002JB001779, 2003.

Castro, R. R., Pacor, F., Puglia, R., Ameri, G., Letort, J., Massa, M. and Luzi, L.: The 2012 May 20 and 29, Emilia earthquakes (Northern Italy) and the main aftershocks: S-wave attenuation, acceleration source functions and site effects, Geophys. J. Int., 195, 597-611, doi: 10.1093/gji/ggt245, 2013.

Convertito, V., Catalli, F. and Emolo, A.: Combining stress transfer and source directivity: the case of the 2012 Emilia seismic sequence, Scientific Reports, 3, 3114, doi: 10.1038/srep03114, 2013.

Gomberg, J., Beeler, N.M. and Blanpied, M.L.: On rate-state and Coulomb failure models. J. Geophys. Res. 105, 7557-7871, 2000.

Harris, R. A.: Introduction to special section: Stress triggers, stress shadows, and implications for seismic hazard, J. Geophys. Res., 103, 24, 347-24, 358, 1998.

Stein, R. S.: The role of stress transfer in earthquake occurrence, Nature, 402, 605–609, 1999.

Stein, R.S., King, G.C.P. and Lin, J.: Change in failure stress on the southern San Andreas fault system caused by the 1992 magnitude = 7.4 Landers earthquake, Science, 258, 1328–1332, 1992.

---

## Author Response (AR1)

**Response to the Referees' Comments**

**M. Dragoni and E. Lorenzano**

Dipartimento di Fisica e Astronomia, Alma Mater Studiorum Università di Bologna, Viale Carlo Berti Pichat 8, 40127 Bologna, Italy

**1 Response to Reviewer 1**

1) The reviewer suggests that we give more details on the 2012 Emilia sequence.

We agree to his request and shall include more details on the data and on their sources in Table 1 and two new figures (numbered as 4 and 5 in the revised version): a map showing the location of the events and a plot of their seismic moments.

2) The reviewer also suggests that we expand the discussion of the model assumptions, in particular assumptions 4, 5 and 6.

These assumptions are suggested by the features of the sequences we are describing, as recorded in seismic catalogues (e.g. Rovida et al., 2011). Assumptions 4 and 5 follow from the observation that sequences are made of

distinct events, each one associated with the failure of a distinct fault in the system, and there is no reactivation of the same fault during a sequence. Assumption 6, stating that the sequence duration is short versus the intersequence time, is justified by seismic history, showing that intersequence intervals are in the order of centuries, while sequence durations are in the order of weeks or months. These comments will be added in section 2.

3) The reviewer finds that the abstract is unclear and favors technical details rather than focusing on the general conclusions of the paper.

The abstract will be rewritten and we shall stress that the key point of the model is to show that the knowledge of the order of activation of faults in a seismic sequence yields information on the state of the fault system before and after the sequence. The concept of permutation is crucial to this aim, because the evolution of the system can be expressed as a sequence of permutations and the order of activation can be described by a particular permutation of the n faults.

4) The reviewer suggests revising the text, particularly the results and conclusions, to make it clear what is new and innovative and/or interesting and significant about this research.

As stated in the Introduction, the aim of the paper is to answer some basic questions concerning seismic sequences. When we observe a sequence, we acknowledge that it is due to a system of n faults that fail one after the other. However, we do not know why the faults fail in that particular order. The order must be a consequence of the initial stress state of the fault system and of the mutual interaction between the faults of the system during the sequence. We show that the knowledge of the order of activation of faults in the sequence yields information on the state of the fault system before and after the sequence. To this aim, we introduce the concept of permutation of the n faults, ranking the faults according to the magnitudes of their Coulomb stresses. Such a permutation describes the state of the system at a given time and changes whenever a fault is activated. The order of activation itself can be described by a particular permutation of the faults. These considerations will be added in the Conclusions of the paper.

5) The reviewer notices that the paper is missing references concerning the subject of fault interaction and triggering.

Following this suggestion, we shall add references to Stein et al. (1992), Harris (1998), Stein (1999), Gomberg et al. (2000), Belardinelli et al. (2003). We shall also add a reference (Love, 1944) for the double-couple point source solution.

6) The reviewer considers that the figures are not detailed well enough, and their motivation or use is not clear: this applies, in particular, to Figure 2.

Figure 2 is a graphical illustration of the evolution of a system made of three faults (n = 3). This case is considered because it can be illustrated graphically, owing to the small number of variables involved. Cases with n ¿ 3 would require higher dimensional spaces. The graphical representation allows a better understanding of the evolution of the state of the fault system during a seismic sequence. This explanation will be added at the beginning of section 7. The captions of figures 2 and 3 will be rewritten and more details will be added in the other captions.

**Response to Reviewer 2**

1-2) The reviewer suggests a discussion of some results obtained by other authors with regard to the 2012 Emilia sequence.

We agree with his request, though the aim of the present work is not to reproduce the details of any particular seismic sequence, but to show how the knowledge of the activation order of faults can give information on the stress state of a fault system. Convertito et al. (2013) suggest that dynamic triggering may have had a role in influencing the seismic sequence, in addition to the variation in permeability and pore-pressure effects due to a massive presence of fluids in the Po Plain basin. We neglected the effect of pore fluid diffusion, on the basis of general considerations in Appendix B, and did not consider dynamic triggering. If these effects are relevant and are introduced

in the calculations, they may alter the sequence of permutations and yield a final permutation different from (64). However, they will not change the general conclusions of the paper. A mention of the possible role of pore fluids will be added in section 2, when the argument is introduced. A short discussion will be added in section 8 and will be recalled at the end of the Conclusions.

3) The reviewer mentions results obtained by Castro et al. (2013) from seismic spectra, suggesting that the events of the Emilia sequence generated relatively low stress drops.

We shall mention this paper in section 8 and show that our model yields values ranging between 0.9 and 1.9 MPa, within the range obtained in that study.

4) As suggested by the reviewer, references will be included in Table 1.

**List of relevant changes made in the manuscript**

1) A more explanatory version of the Abstract (p.1, lines 1–13);

2) An expanded discussion of the model assumptions (p.2, lines 27–32; p.3, lines 1–2);

3) A mention of the possible role of pore fluids in the evolution of a seismic sequence (p.5, lines 18–19);

4) Reasons for the application of the model to a system made of three faults (p.11, lines 7–10);

5) Discussion of the role of dynamic triggering and pore fluids diffusion on the evolution of the 2012 Emilia sequence (p.13, lines 22–31);

6) Estimate of the stress drops associated with the events of the 2012 Emilia sequence (p.14, lines 15–17);

7) A more detailed version of the Conclusions (p.15, lines 10–23; p.16, lines 14–15);

8) Additional references about fault interaction and the double-couple point source solution (p.19, lines 3, 20, 21, 25; p.20, line 19);

9) A more explanatory version of the Figure Captions (p.20, lines 28–34; p.21, lines 1–10);

10) References and additional data in Table 1 (p.22);

11) Location of the events of the 2012 Emilia sequence and the associated

seismic moments (p.25, Figure 4; p.26, Figure 5).

**References**

[revised manuscript text omitted]